# Efficacy and Safety of Transvenous Lead Extraction at the Time of Upgrade from Pacemakers to Cardioverter-Defibrillators and Cardiac Resynchronization Therapy

**DOI:** 10.3390/ijerph20010291

**Published:** 2022-12-24

**Authors:** Paweł Stefańczyk, Dorota Nowosielecka, Anna Polewczyk, Wojciech Jacheć, Andrzej Głowniak, Jarosław Kosior, Andrzej Kutarski

**Affiliations:** 1Department of Cardiology, Pope John Paul II Province Hospital, 22-400 Zamość, Poland; 2Department of Cardiac Surgery, Pope John Paul II Province Hospital, 22-400 Zamość, Poland; 3Department of Physiology, Pathophysiology and Clinical Immunology, Institute of Medical Sciences, Jan Kochanowski University, 25-369 Kielce, Poland; 4Department of Cardiac Surgery, Świętokrzyskie Cardiology Center, 25-736 Kielce, Poland; 52nd Department of Cardiology, Faculty of Medical Sciences in Zabrze, Silesian Medical University in Katowice, 41-800 Zabrze, Poland; 6Department of Cardiology, Medical University, 20-059 Lublin, Poland; 7Department of Cardiology, Masovian Specialist Hospital, 26-617 Radom, Poland

**Keywords:** device upgrade, transvenous lead extraction, prophylaxis of lead abandonment, restoration of venous access

## Abstract

Background: Upgrading from pacemakers to ICDs and CRTs is a difficult procedure, and often, transvenous lead extraction (TLE) is necessary for venous access. TLE is considered riskier in patients with multiple diseases. We aimed to assess the complexity, risk, and outcome of TLE among CRT and ICD candidates. Methods: We analyzed clinical data from 2408 patients undergoing TLE between 2006 and 2021. There were 138 patients upgraded to CRT-D, 33 patients upgraded to CRT-P and 89 individuals upgraded to ICD versus 2148 patients undergoing TLE for other non-infectious indications. Results: The need for an upgrade was the leading indication for TLE in only 36–66% of patients. In 42.0–57.6% of patients, the upgrade procedure could be successfully done only after reestablishing access to the occluded vein. All leads were extracted in 68.1–76.4% of patients, functional leads were retained in 20.2–31.9%, non-functional leads were left in place in 0.0–1.1%, and non-functional superfluous leads were extracted in 3.6–8.4%. The long-term survival rate of patients in the CRT-upgrade group was lower (63.8%) than in the non-upgrade group (75.2%). Conclusions: Upgrading a patient from an existing pacemaker to an ICD/CRT is feasible in 100% of cases, provided that TLE is performed for venous access. Major complications of TLE at the time of device upgrade are rare and, if present do not result in death.

## 1. Introduction

Changes in the health status of patients with cardiac implantable electronic devices (CIED) sometimes require changes in pacing mode (from AAI or VVI to DDD, from any type of pacing to an ICD, from ICD-V to ICD-D, or from any pacing mode to CRT-P or CRT-D) [1,2,3,4,5]. In patients without lead-related venous obstruction, insertion of any additional pacing lead (atrial, right, or left ventricular) is not a problem. In contrast, the need to implant a high voltage (HV) lead requires consideration of removing the previously implanted ventricular lead. In patients with lead-related venous obstruction, according to the guidelines, it is possible to extract ventricular leads, restore venous access, and implant ICD (and/or CS) leads, or (worse, but unfortunately still an accepted choice) place a new CIED on the opposite side of the chest, leaving the old lead in place (2b class indication for lead extraction). If contralateral implantation is not feasible due to venous obstruction or the presence of a catheter, there are class 1 or 2a indications for lead replacement [6,7]. There are several studies on lead extraction at the time of upgrade, but all of them focus on reestablishing access to an occluded implant vein [8,9,10,11,12]. Lead-related venous stenosis is a frequent finding in CIED carriers (patent veins are found in 17.2% of patients, mild stenosis in 19.7%, moderate in 20.8%, severe in 20.0%, and complete obstruction in 22.3% of patients) [13,14]. Therefore, in more than 50% of patients, we should be aware of the difficulties or inability to implant a new lead. [15]. Patency of the access vein is even more important for a successful upgrade than for ordinary lead insertion due to a larger diameter of introducers and more leads for simultaneous implantation. On the other hand, avoidance of lead abandonment is preferred to reduce the risk of lead-to-lead scarring [16,17] and to prevent pocket bulging caused by redundant lead loops. To the best of our knowledge, no previous study has investigated a broad range of indications for TLE at the time of an upgrade to an ICD or CRT and/or the safety and effectiveness of lead extraction in such patients.

## 2. Methods

### 2.1. Study Population

All procedures of transvenous lead extraction performed from March 2006 to July 2021 by the same primary operator at three high volume centers were analyzed. Patient and lead data (patient demographics, comorbidities, device type and history, preoperative venous patency, TLE effectiveness, major complications, and short- and long-term mortality) were entered into the computer on an ongoing basis. Patients with infectious indications for lead removal were excluded from the study. The study population was divided according to device upgrade type: group 1a: upgrade to CRT-D (n = 138), group 1b: upgrade to CRT-P (n = 33), and group 1c: upgrade to ICD (n = 89). All the upgrade groups (CRT-D, CRT-P, ICD-V, and ICD-D) made up group 1, which consisted of 260 patients. The remaining patients undergoing TLE for other non-infectious indications without upgrades to ICD and CRT were assigned to the control group, i.e., group 2 (n = 2148). We analyzed and compared clinical and procedural data, the effectiveness and safety of TLE, and long-term mortality in individual patient groups.

### 2.2. Lead Extraction Procedure

Indications for TLE, procedure effectiveness, and complications were assessed according to the 2009 and 2017 HRS consensus and the 2018 EHRA guidelines [6,7,18]. The efficacy of TLE was determined based on the percentage of procedural success, and clinical success including complete and partial radiographic success. Procedural success was defined as the removal of all targeted leads and lead material from the vascular space with the absence of any permanently disabling complication or procedure-related death. Clinical success was defined as the removal of all targeted leads or the retention of a small portion (<4 cm) of the lead that did not negatively impact the outcome goals of the procedure (i.e., residual lead did not increase the risk of perforation, embolic events, perpetuation of infection, or cause any undesired outcome), the absence of any permanently disabling complication, or a procedure-related death.

The complications of TLE were defined as major complications that were life-threatening, resulted in significant or permanent disability or death, or required surgical intervention [6,7,18].

Antibiotic prophylaxis in all patients consisted of a first-generation cephalosporin administered as a bolus an hour before TLE. In most procedures, standard stylets were used to stiffen the leads. Locking stylets (Liberator Locking Stylet, Cook Medical Inc., Bloomington, IN, USA) were used only for the extraction of the oldest leads when the estimated risk of lead break was high. Simple traction or traction on a locking stylet with insulation-bound suture was very rarely applied as our intention was to preserve or restore venous access for the implantation of a new lead(s) (Figure 1). Lead extraction was usually performed using non-powered mechanical telescoping polypropylene sheaths (Byrd Dilator Sheaths, Cook Medical Inc., Bloomington, IN, USA) of all diameters and lengths, and various stylets. When the polypropylene telescoping sheaths appeared ineffective, powered mechanical sheath systems (Evolution Mechanical Dilator Sheath, Cook Medical Inc., Bloomington, IN, USA; TightRail Rotating Dilator Sheath, Spectranetics, Colorado Springs, CO, USA) were used. A combination of approaches, using two or more different (jugular, subclavian, or femoral) access sites, was selected when conventional methods were insufficient. Laser and electrosurgical dissection sheaths were not used.

In this study, extractions were performed according to different organizational models (spanning 16 years of experience), ranging from procedures in the electrophysiology laboratory using intravenous analgesia/sedation [19] to procedures in the hybrid room under general anesthesia. Over the past six years, the core extraction team has consisted of the same highly experienced TLE operator, experienced echocardiographer, and dedicated cardiac surgeon [20,21,22].

Continuous transesophageal echocardiography (TEE) for monitoring patients undergoing TLE has become routine in the past six years. Philips iE33 or GE Vivid S 70 machines equipped with X7-2t Live 3D or 6VT-D probes were used. Intraprocedural TEE provided an opportunity to check the process of pulling on cardiac walls and right ventricular caving inward and to detect a drop in systolic blood pressure in response to this maneuver, thus facilitating early recognition of TLE complications. TEE at the time of new lead implantation allowed for a more accurate assessment of tip location, whereas during left ventricular lead implantation, TEE helped in the navigation of the delivery catheter into the CS ostium. TEE monitoring was used during 1123 TLEs for non-infectious indications, and TEE navigation assisted in locating the CS ostium in 76 of 138 upgrades to CRT-D (54.3%) and in 16 of 33 upgrades to CRT-P (48.5%).

Venous patency before lead extraction is defined as maximal narrowing and the number of affected veins. 

### 2.3. Dataset and Statistical Methods

The Shapiro-Wilk test showed that most continuous variables were normally distributed. For uniformity, all continuous variables are expressed as mean ± standard deviation. Categorical variables are presented as numbers and percentages. The study population was divided into groups: 1a—patients upgraded from pacemaker or ICD (VR or DR) to CRT-D; 1b—patients upgraded from pacemaker to CRT-P; and 1c—patients upgraded from pacemaker to ICD (VR or DR); group 1—comprising patients from groups 1a-1c; and group 2—patients who underwent TLE for other indications without upgrades and serving as a comparison group. The significance of differences between groups (1, 1a, 1b, and 1c vs. 2) was determined using the nonparametric Chi^2^ test with Yates correction or the unpaired Mann-Whitney U test, as appropriate. A p-value less than 0.05 was considered statistically significant. Data analysis was performed using Statistica 13.3 (TIBCO Software Inc., Palo Alto, CA, USA).

### 2.4. Approval of the Bioethics Committee

All patients gave their informed written consent to undergo TLE and use anonymous data from their medical records, as approved by the Bioethics Committee at the Regional Chamber of Physicians in Lublin, no. 288/2018/KB/VII. The study was carried out in accordance with the ethical standards of the 1964 Declaration of Helsinki.

## 3. Results

A total of 2408 patients (56.6% male), with a mean age of 64.9 ± 16.3 (range 5–96 years), underwent lead extraction. Upgrade patients were more likely to have heart failure and renal failure, and there were significantly fewer women in the entire upgrade group. Patients upgraded to an ICD were younger and less often presented with severe heart failure and reduced left ventricular ejection fraction (LVEF) compared to those upgraded to a CRT. Patients with severe heart failure and significantly reduced LVEF were most likely to undergo an upgrade procedure to CRT-D. Severe renal failure was most common in patients upgraded to CRT-P, but the highest Charlson comorbidity index was noted in the CRT-D upgrade group.

Analysis of indications for TLE in individual groups showed high heterogeneity related to the coexistence of two and sometimes even three main indications (e.g., lead failure and venous obstruction with the need for a CIED upgrade). Less than 40% of patients in the entire upgrade group had a different, more important indication for TLE, and upgrading was done at the time of a CIED-related procedure. In most patients, the leading indication for TLE was an intention to upgrade existing devices (Table 1).

Analysis of venograms obtained before lead extraction showed a similar degree of narrowing: moderate stenosis in 19.8–21.9% of patients, severe stenosis in 15.5–25.0%, and complete occlusion in 20.7–23.3%. The extent of venous obstruction, considered as the number of affected veins with moderate/severe stenosis and complete occlusion, was also similar in all the compared groups. This means that unavoidable lead-related venous obstruction had no influence on lead management strategy, and a contralateral implant was not necessary in any of the cases.

In this study, the groups did not differ significantly in system-related risk factors such as abandoned leads, number of leads in the heart (4 and >4 leads in the heart) and leads on both sides of the chest. Only the number of preceding CIED-related procedures was significantly lower in patients who were upgraded to CRT-D.

Risk factors for major complications of TLE and procedure complexity were evenly distributed between the groups, but the calculated risk of major complications using the SAFeTY-TLE score [23] (sum of assigned points and probability expressed as a percentage) was highest in the CRT-P upgrade group (longer implant duration, more previous CIED-related procedures, more female patients) (Table 2).

A comparison of TLE complexity revealed prolonged procedure duration (skin-to-skin time) in patients who were upgraded to CRT-D or CRT-P, as additional time was needed for the implantation of left ventricular leads. Only the duration of lead extraction measured as time for removal of all leads (sheath-to-sheath time) and mean time of single lead extraction were similar in all study groups. Procedure complexity, defined as the need to use alternative access sites, lead-to-lead scarring, Byrd dilator collapse/torsion, and the need to use second-line tools (Evolution (old and new) or TightRail, metal sheaths, and lasso catheters/snares), was similar in all study groups. Only a lead fracture during extraction was significantly more common in patients who underwent an upgrade procedure to ICD.

Analysis of the extraction approach showed that in 42.0–57.6% of patients, only reestablishing access to the occluded vein permitted a successful upgrade. In most of the remaining patients, venous access was maintained (one, two, or even three guide wires passed through the lead extraction sheath and new introducers). In a small percentage of patients, due to the previously used parasternal approach, the operator tried to perform a more peripheral vein puncture to prevent crush syndrome (Table 3).

The goal of lead management in patients undergoing system upgrades was “no lead left behind” on completion of TLE. All leads were extracted in 68.1–76.4% of patients; functional leads were left in place in 20.2–31.9% of patients; non-functional leads were left behind in only 0.0–1.1% of patients; and non-functional superfluous leads were extracted in 3.6–8.4% of patients.

Evaluation of TLE efficacy and safety showed zero occurrences of major complications and tricuspid valve damage during TLE, no need for emergent cardiac surgery, and no procedure-related deaths (intra- and post-procedural) in all upgrade groups. The rate of clinical success in the upgrade groups was 100%. Due to the lower occurrence of partial radiographic success, the rate of procedural success was higher in upgraded patients (98.1% vs. 95.1% in patients undergoing TLE for other indications; *p* = 0.042).

Analysis of short-, medium-, and long-term prognosis after TLE showed that survival in the group upgraded to CRT was lower (63.8%) than in patients without the upgrade (75.2%), *p <* 0.001, but no association with 48-h or 1-month mortality by device type was demonstrated. However, mortality at more than one year and three years after TLE was significantly higher among patients with the CRT upgrade (8.1 and 12.8%) than the non-upgrade control group (4.4 and 7.5%) (Table 4).

## 4. Discussion

The purpose of lead extraction at the time of CIED upgrade is several fold: to prevent superfluous lead abandonment, prevent abandonment of dysfunctional leads, reestablish access to an occluded vein, and, rarely, remove previously abandoned leads. Current guidelines for transvenous lead extraction recommend removing unnecessary leads in patients with long life expectancies [6,7,18]. There is still controversy over lead management strategy (including lead abandonment), and the final opinion of very experienced high-volume operators is well-balanced [24,25,26,27,28,29,30,31,32,33]. Conclusions from these reports gently promote the avoidance of lead abandonment, provided that all feasible precautions are taken to ensure safety. This study in a large population of patients undergoing TLE during an upgrade procedure to ICD or CRT demonstrated that all leads were successfully extracted in 68.1–76.4% of cases and non-functional leads were left behind in only 0.0–1.1% of cases. This strategy was chosen based on research findings suggesting that lead abandonment is often associated with increased risk of complications [24], abandoned leads complicate the management of cardiac device infections, resulting in worse clinical outcomes [25], the extraction of abandoned leads is associated with a lower risk of device infections at 5 years [26], and patients with abandoned leads are more likely to require laser extraction [27]. Additionally, abandoned leads increase procedural complexity, including a higher rate of bailout femoral extraction, which may be associated with lower clinical success [28], whereas removal of non-infectious superfluous leads seems to be a safe and effective therapeutic option [32].

The analysis of TLE efficacy in patients undergoing upgrade to ICD and CRT in this study showed very high effectiveness and safety of the procedure: 100% complete clinical success rate, 98% complete procedural success rate, and no major complications in this patient population. To date, there are few studies that have investigated lead extraction in patients undergoing system upgrades. Analysis of laser lead extraction performed in 71 patients scheduled for device upgrade/revision who had occluded or functionally obstructed venous anatomy also showed high efficiency (procedural success in 100% of patients, 3% major complications) [9]. Similarly, in a case series reported by Bracke et al. [34] and Gula et al. [35], the efficiency of lead extraction was 100 percent, and no major complications were noted.

Consideration should also be given to the additional use of transesophageal echocardiography (TEE) when inserting new leads after TLE. TEE facilitates the monitoring of the intubation process of the coronary sinus, thus shortening the time of fluoroscopy, and if high-voltage lead is required, TEE can help accurately assess the relationship between the proximal coil end and the level of the valve and leaflets (Figure 2).

One of the most important issues in patients undergoing system upgrades is venous patency and the need to have enough space for two large introducers, which may necessitate the extraction of the existing lead. Imaging with venography is mandatory before any attempt at a system upgrade [13,14,15]. In this study, like previous reports [9,13,14,15], significant lead-related venous obstruction was found in about 60% of upgrade patients (Figure 3). Despite difficult venous access in these patients, there was no case in which a contralateral implant was necessary.

TEE monitoring during lead extraction procedures has become the standard clinical approach whose importance cannot be overestimated. In patients undergoing TLE with subsequent upgrade it allows for a more accurate assessment of the location of RV lead tips, and if implantation of CS lead for left ventricular pacing is required, TEE is used for navigation during insertion of the delivery catheter into the CS ostium (Figure 2) [27,36,37].

The present study also evaluated the survival of patients undergoing lead extraction. It is worth highlighting that there were no periprocedural deaths in patients who had undergone system upgrade. Worse long-term outcomes can be accounted for by the poorer clinical characteristics of patients upgraded to ICD/CRT.

## 5. Conclusions

Upgrading a patient from an existing pacemaker to an ICD/CRT-P/CRT-D is feasible in 100% of cases, provided that transvenous lead extraction is performed for venous access, if necessary.In clinical practice, most upgrades from PM to ICD/CRT coincide with system revision/lead extraction due to lead failure. A separate indication for upgrading is less frequent (under 60%).Lead extraction at the time of upgrade prevents superfluous lead abandonment.Major complications of lead extraction for device upgrades are rare (0% at experienced centers) and, if present, do not result in death.

### Study Limitation

All procedures in this study were performed by the same very experienced operator at three high-volume centers; therefore, it may be difficult to reproduce the results in centers with lower volumes and less expertise. A mechanical-only approach to lead extraction was used throughout the study.

## Figures and Tables

**Figure 1 ijerph-20-00291-f001:**
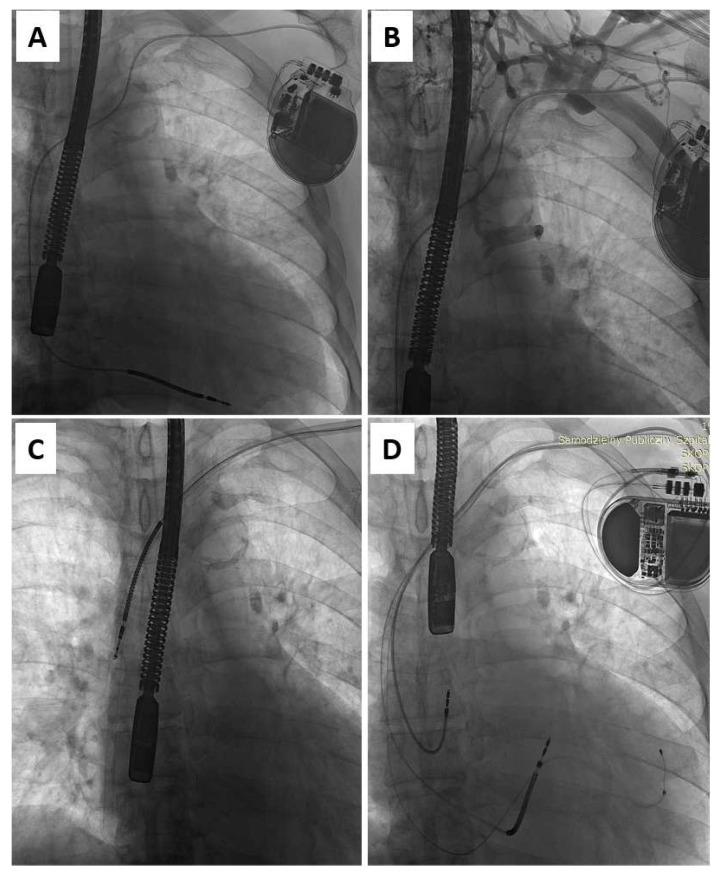
Upgrade from ICD-V (**A**) to CRT-D in the patient undergoing TLE for lead-related venous obstruction (**B**). Moving the introducer sheath over the lead to the heart facilitates not only the removal of the lead (**C**) but also reestablishing access for the implantation of three new leads (**D**).

**Figure 2 ijerph-20-00291-f002:**
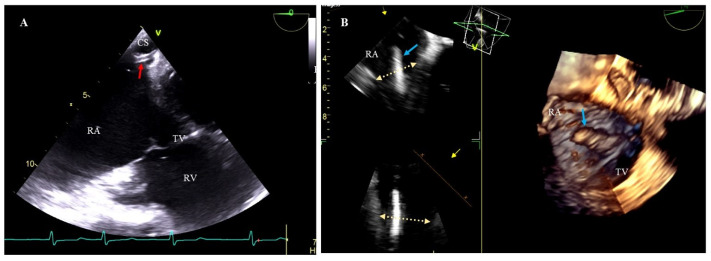
Additional use of TEE at new lead implantation during TLE. (**A**) 2D TEE images- lower esophageal view. A catheter (red arrow) was inserted into the coronary sinus (CS) under X-ray and TEE guidance. (**B**) 3D TEE images—mid-esophageal view. The coil (blue arrow) protrudes into RA above the tricuspid valve level (yellow arrows). RA—right atrium, RV—right ventricle, TV—tricuspid valve.

**Figure 3 ijerph-20-00291-f003:**
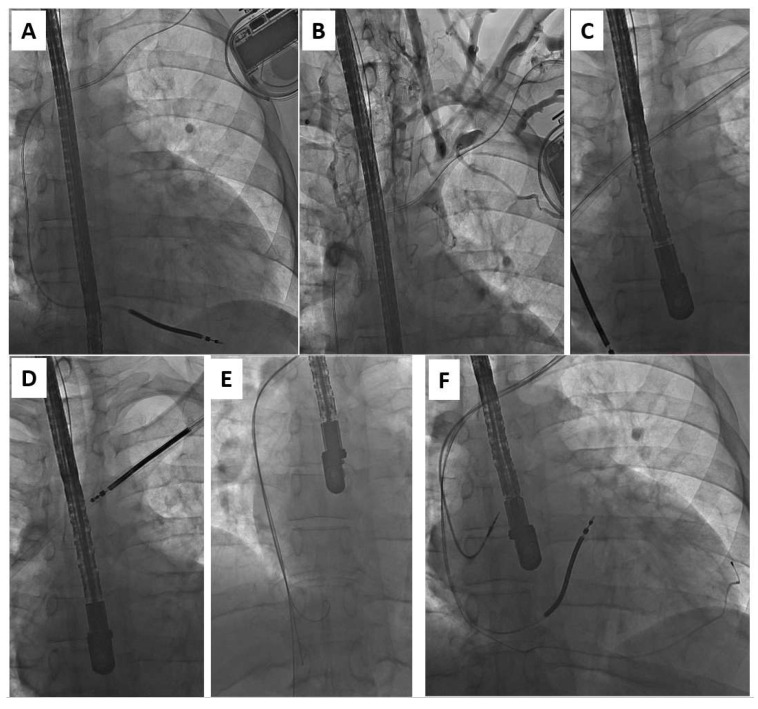
Upgrade from ICD-V (**A**) to CRT-D in the patient with lead-related venous obstruction (**B**). Moving the preparation sheath over the lead to the heart allows not only for lead removal (**C**,**D**) but also reestablishing venous access (**E**) for the implantation of three new leads (**F**).

**Table 1 ijerph-20-00291-t001:** Patient characteristics and indications for lead extraction.

		Upgrade to CRT-D	Upgrade to CRT-P	Upgrade to ICD	All Upgrades (to CRT-D, CRT-P and to ICD-V or ICD-D)	TLE for Other Non-Infectious Indications without Upgrades
**Group**		**1a**	**1b**	**1c**	**1**	**2**
**Number of patients**		**138**	**33**	**89**	**260**	**2148**
		**Mean ± SD** **n (%)**	**Mean ± SD** **n (%)**	**Mean ± SD** **n (%)**	**Mean ± SD** **n (%)**	**Mean ± SD** **n (%)**
Patient age at TLE	[years]	68.35 ± 10.57 *p* = 0.149	70.70 ± 11.41 *p* = 0.400	63.19 ± 14.66 *p* = 0.039	66.88 ± 12.49 *p* = 0.555	64.69 ± 16.73
Patient age at first device implantation	[years]	60.51 ± 11.69 *p* = 0.058	60.85 ± 14.78 *p* = 0.215	53.54 ± 16.62 *p* = 0.015	58.17 ± 14.30 *p* = 0.673	56.02 ± 18.27
Sex (% of female patients)	n (%)	21 (15.22) *p <* 0.001	16 (48.48) *p* = 0.880	27 (30.34) *p* = 0.006	64 (24.62) *p <* 0.001	980 (45.62)
Underlying heart disease: IHD	n (%)	90 (67.22) *p* = 0.031	16 (48.48) *p* = 0.538	53 (59.55) *P =* 0.507	159 (61.15) *p* = 0.089	1190 (54.00)
NYHA III and IV	n (%)	78 (56.52) *p <* 0.001	18 (54.55) *p* = 0.938	21 (23.60) *p <* 0.001	117 (45.00) *p* = 0.002	212 (9.870)
Congestive heart failure (symptomatic before TLE)	n (%)	98 (71.01) *p <* 0.001	17 (51.52) *p* = 0.789	33 (37.08) *p <* 0.001	148 (56.92) *p* = 0.689	302 (14.06)
LVEF average	[%]	28.23 ± 7.96 *p <* 0.001	35.64 ± 10.32 *p <* 0.001	39.46 ± 15.39 *p <* 0.001	33.03 ± 12.44 *p <* 0.001	51.87 ± 14.39
Normal LVEF (>50%)	n (%)	2 (1.460) *p <* 0.001	3 (9.090) *p <* 0.001	24 (26.97) *p <* 0.001	29 (11.20) *p <* 0.001	1286 (60.07)
LVEF mildly reduced (41–50%)	n (%)	4 (2.920) *p <* 0.001	3 (9.090) *p <* 0.001	8 (8.990) *p <* 0.001	15 (5.790) *p <* 0.001	330 (15.41)
LVEF moderately reduced (30–40%)	n (%)	53 (38.69) *p <* 0.001	19 (57.58) *p <* 0.001	34 (38.20) *p <* 0.001	106 (40.93) *p <* 0.001	339 (15.83)
LVEF significantly reduced (<30%)	n (%)	78 (56.93) *p <* 0.001	8 (24.24) *p* = 0.005	23 (25.84) *p <* 0.001	109 (42.08) *p <* 0.001	186 (8.69)
Renal failure (any)	n (%)	46 (33.33) *p <* 0.001	10 (30.30) *p* = 0.068	22 (24.72) *p* = 0.070	78 (30.00) *p <* 0.001	360 (16.76)
Renal failure: severe or hemodialysis (creat. 2.3 and >2.3 mg/dL)	n (%)	6 (4.350) *p* = 0.559	10 (30.30) *p <* 0.001	4 (4.490) *p* = 0.304	10 (3.850) *p <* 0.001	48 (2.250)
Renal failure: moderate (create. 1.3–2.2 mg/dL)	n (%)	40 (28.99) *p <* 0.001	0 (0.00) *p <* 0.001	18 (20.22) *p* = 0.183	68 (26.15) *p <* 0.001	312 (14.53)
Diabetes	n (%)	40 (28.99) *p <* 0.001	4 (12.12) *p* = 0.324	16 (17.98) *p* = 0.633	60 (23.08) *p <* 0.001	369 (17.18)
Charlson comorbidity index	[points]	6.53 ± 3.82 *p <* 0.001	4.58 ± 3.02 *p* = 0.374	4.701 ± 4.11 *p* = 0.795	5.66 ± 3.93 *p <* 0.001	4.31 ± 3.52
**Indications for TLE in the study population** **(primary or secondary/predominant or accompanying)**						
Mechanical lead damage (electrical failure)	n (%)	9 (6.52) *p <* 0.001	4 (12.12) *p <* 0.001	11 (12.36) *p <* 0.001	24 (9.32) *p <* 0.001	921 (42.88)
Lead dysfunction (exit/entry block, dislodgement, extracardiac pacing)	n (%)	6 (4.35) *p <* 0.001	4 (12.12) *p <* 0.001	5 (5.62) *p <* 0.001	15 (5.77) *p <* 0.001	414 (19.27)
Lead dysfunction caused by (usually dry) perforation	n (%)	4 (2.90) *p <* 0.001	1 (3.03) *p <* 0.001	2 (2.25) *p <* 0.001	7 (2.69) *p <* 0.001	375 (17.46)
Change of pacing mode/upgrading, downgrading	n (%)	92 (66.67) *p <* 0.001	12 (36.36) *p <* 0.001	57 (64.04) *p <* 0.001	161 (61.92) *p <* 0.001	48 (2.23)
Abandoned lead/prevention of abandonment (AF, multiple leads)	n (%)	2 (1.45) *p* = 0.149	2 (6.06) *p* = 0.968	2 (2.25) *p* = 0.481	6 (2.31) *p* = 0.157	94 (4.38)
Threatening/potentially threatening lead (loops, free endings, left heart, LDTVD)	n (%)	1 (0.72) *p* = 0.046	6 (18.18) *p* = 0.015	3 (3.37) *p* = 0.239	10 (3.85) *p* = 0.005	103 (4.80)
Other (MRI indications, cancer, painful pocket, pacemaker/ICD no longer needed)	n (%)	3 (2.17) *p* = 0.306	0 (0.00) *p* = 0.426	0 (0.00) *p* = 0.081	3 (1.15) *p* = 0.020	94 (4.38)
Reestablishing venous access (symptomatic occlusion, SVC syndrome, lead replacement/upgrading)	n (%)	21 (15.22) *p <* 0.001	4 (12.12) *p <* 0.001	9 (10.11) *p <* 0.001	34 (13.08) *p <* 0.001	97 (4.52)

TLE—transvenous lead extraction, CRTD—cardiac resynchronization cardioverter defibrillator, CRT-P—cardiac resynchronization pacemaker, ICD—implantable cardioverter defibrillator, ICD-V—one chamber ICD, ICD-D—dual chamber ICD, SD—standard deviation, N/[n]—number, IHD—ischemic heart disease, NYHA—New York Heart Association functional class, LVEF—left ventricle ejection fraction, AF—atrial fibrillation, LDTVD—lead-derived tricuspid valve defect, MRI—magnetic resonance imaging, SVC—superior vena cava. The control group 2 served as a comparison group in the statistical evaluation.

**Table 2 ijerph-20-00291-t002:** Venous patency before lead extraction (maximal narrowing and number of affected veins) and risk factors for TLE complications and procedure complexity.

		Upgrade to CRT-D	Upgrade to CRT-P	Upgrade to ICD	All Upgrades (to CRT-D, CRT-P and to ICD-V or ICD-D)	TLE for Other Non-Infectious Indications without Upgrades
**Group**		**1a**	**1b**	**1c**	**1**	**2**
**Number of Patients**		**138**	**33**	**89**	**260**	**2148**
		**n (%)** **Mean ± SD**	**n (%)** **Mean ± SD**	**n (%)** **Mean ± SD**	**n (%)** **Mean ± SD**	**n (%)** **Mean ± SD**
**Venous patency before lead extraction (maximal narrowing)**						
Patent (no visible stenosis)	n (%)	21 (18.10) *p* = 0.975	4 (14.29) *p* = 0.771	17 (20.73) *p* = 0.443	42 (18.58) *p* = 0.855	333 (17.54)
Mild stenosis (<1/3 decrease in the vein diameter)	n (%)	30 (25.86) *p* = 0.410	4 (14.29) *p* = 0.474	15 (18.29) *p* = 0.795	49 (21.68) *p* = 0.968	398 (20.97)
Moderate stenosis (1/3 to 2/3 decrease in the vein diameter)	n (%)	23 (19.83) *p* = 0.599	6 (21.43) *p* = 0.900	18 (21.95) *p* = 0.853	47 (20.80) *p* = 0.826	405 (21.34)
Severe stenosis (≥2/3 decrease in the vein diameter, but still patent)	n (%)	18 (15.52) *p* = 0.151	7 (25.00) *p* = 0.837	13 (15.85) *p* = 0.459	38 (16.81) *p* = 0.171	386 (20.34)
Complete occlusion	n (%)	24 (20.69) *p* = 0.974	7 (25.00) *p* = 0.745	19 (23.17) *p* = 0.767	50 (22.12) *p* = 0.547	376 (19.81
Venograms were not obtained (contraindications and technical problems)	n (%)	22 (15.94) *p* = 0.014	5 (15.15) *p* = 0.691	7 (7.865) *p* = 0.390	34 (13.08) *p* = 0.484	250 (11.64)
**Venous patency before lead extraction (number of affected veins)**						
Mild narrowing	n (%)	51 (43.97) *p* = 0.557	8 (28.57) *p* = 0.315	32 (39.02) *p* = 0.808	91 (40.27) *p* = 0.832	733 (38.50)
One vein significantly affected (moderate/severe stenosis or complete occlusion)	n (%)	39 (33.62) *p* = 0.762	9 (32.14) *p* = 0.896	32 (39.02) *p* = 0.264	80 (35.40) *p* = 0.814	755 (39.65)
Two veins are significantly affected (moderate/severe stenosis or complete occlusion)	n (%)	24 (20.69) *p* = 0.913	11 (39.29) *p* = 0.032	16 (19.51) *p* = 0.996	51 (22.57) *p* = 0.427	374 (19.64)
Three veins are significantly affected (moderate/severe stenosis or complete occlusion)	n (%)	2 (1.72) *p* = 0.782	0 (0.00) *p* = 0.999	2 (2.44) *p* = 0.925	4 (1.77) *p* = 0.792	33 (1.73)
Four veins are significantly affected (moderate/severe stenosis or complete occlusion)	n (%)	0 (0.00) *p* = 0.952	0 (0.00) *p* = 0.320	0 (0.00) *p* = 0.808	0 (0.00) *p* = 0.612	9 (0.37)
Venograms were not obtained (contraindications, technical problems) or lack of an appropriate assessment	n (%)	22 *p* = 0.136	5 *p* = 0.686	7 *p* = 0.394	34 *p* = 0.474	244
**Risk factors related to pacing history**						
Abandoned leads before TLE	n (%)	6 (4.35) *p* = 0.076	2 (6.06) *p* = 0.756	11 (12.36) *p* = 0.407	19 (7.31) *p* = 0.380	197 (9.17)
Number of leads in the heart before TLE	[n]	1.88 ± 0.68 *p* = 0.986	1.99 ± 0.66 *p* = 0.718	1.86 ± 0.74 *p* = 0.522	1.88 ± 0.70 *p* = 0.807	1.89 ± 0.72
4 and >4 leads in the heart before TLE	n (%)	4 (2.90) *p* = 0.682	1 (3.03) *p* = 0.836	3 (3.37) *p* = 0.610	8 (3.08) *p* = 0.363	43 (2.00)
Leads on both sides of the chest before TLE	n (%)	2 (1.45) *p* = 0.951	1 (3.03) *p* = 0.863	3 (3.37) *p* = 0.559	6 (2.31) *p* = 0.840	41 (1.91)
Number of procedures before TLE	[n]	1.59 ± 0.74 *p* = 0.036	2.10 ± 1.32 *p* = 0.181	1.738 ± 1.17 *p* = 0.785	1.174 ± 1.02 *p* = 0.235	1.72 ± 0.95
Time since last CIED procedure (any) (months)	[months]	49.25 ± 31.20 *p* = 0.417	59.05 ± 43.83 *p* = 0.930	64.17 ± 35.89 *p* = 0.016	56.86 ± 35.42 *p* = 0.051	50.55 ± 37.25
**Procedure-related risk factors for major complications and procedure complexity**						
Number of extracted leads per patient	[n]	1.46 ± 0.61 *p* = 0.985	1.61 ± 0.61 *p* = 0.188	1.63 ± 0.76 *p* = 0.164	1.54 ± 0.66 *p* = 0.224	1.48 ± 0.65
Three or more leads extracted	n (%)	6 (4.35) *p* = 0.346	2 (9.06) *p* = 0.868	9 (10.11) *p* = 0.320	17 (6.54) *p* = 0.979	146 (6.80)
Approach other than lead implant vein	n (%)	0 (0.00) *p* = 0.105	1 (3.03) *p* = 0.696	2 (2.25) *p* = 0.919	3 (1.15) *p* = 0.267	53 (2.47)
Extraction of abandoned lead(s) (any)	n (%)	5 (3.62) *p* = 0.065	2 (6.06) *p* = 0.861	10 (11.24) *p* = 0.468	17 (6.54) *p* = 0.348	177 (8.24)
Oldest extracted lead per patient	[years]	7.80 ± 5.45 *p* = 0.130	9.84 ± 7.06 *p* = 0.071	9.27 ± 4.81 *p* = 0.085	8.57 ± 5.98 *p* = 0.622	8.54 ± 6.36
Average lead dwell time per patient	[years]	7.51 ± 4.95 *p* = 0.152	9.52 ± 6.49 *p* = 0.058	8.43 ± 4.76 *p* = 0.112	8.08 ± 5.15 *p* = 0.612	8.20 ± 5.87
Average lead implant duration per group	[years]	8.21 ± 3.54 *p* = 0.333	10.22 ± 7.35 *p* = 0.076	9.03 ± 5.83 *p* = 0.058	8.77 ± 5.93 *p* = 0.331	8.79 ± 6.23
Cumulative lead dwell time per patient	[years]	12.25 ± 11.84 *p* = 0.543	16.41 ± 14.83 *p* = 0.084	14.77 ± 12.80 *p* = 0.066	13.62 ± 12.62 *p* = 0.238	13.14 ± 12.65
SAFeTY TLE score for risk of MC [points]	[points]	3.07 ± 3.66 *p <* 0.001	7.38 ± 4.78 *p* = 0.045	5.42 ± 4.28 *p* = 0.501	4.94 ± 4.17 *p <* 0.001	5.67 ± 4.14
SAFeTY TLE score for risk of MC [%]	[%]	0.96 ± 1.61 *p <* 0.001	2.58 ± 3.23 *p* = 0.233	1.76 ± 4.84 *p* = 0.804	1.44 ± 3.28 *p <* 0.001	1.61 ± 2.65

TLE—transvenous lead extraction, CRTD—cardiac resynchronization cardioverter defibrillator, CRT-P—cardiac resynchronization pacemaker, ICD—implantable cardioverter defibrillator, ICD-V—one chamber ICD, ICD-D—dual chamber ICD, SD—standard deviation, N/[n]—number, MC—major complications. The control group 2 served as a comparison group in the statistical evaluation.

**Table 3 ijerph-20-00291-t003:** Extraction procedure complexity.

		Upgrade to CRT-D	Upgrade to CRT-P	Upgrade to ICD	All Upgrades (to CRT-D, CRT-P and to ICD-V or ICD-D)	TLE for Other Non-Infectious Indications without Upgrades
**Group**		**1a**	**1b**	**1c**	**1**	**2**
**Number of Patients**		**138**	**33**	**89**	**260**	**2148**
		**n (%)** **Mean ± SD**	**n (%)** **Mean ± SD**	**n (%)** **Mean ± SD**	**n (%)** **Mean ± SD**	**n (%)** **Mean ± SD**
**Procedure complexity and outcomes**						
Procedure duration (skin-to-skin)	[minutes]	83.62 ± 21.88 *p <* 0.001	79.55 ± 17.52 *p <* 0.001	65.60 ± 23.12 *p* = 0.550	76.93 ± 23.27 *p <* 0.001	64.19 ± 24.27
procedure duration (sheath-to-sheath)	[minutes]	12.50 ± 17.81 *p* = 0.250	12.19 ± 11.07 *p* = 0.489	16.19 ± 23.77 *p* = 0.804	13.72 ± 19.45 *p* = 0.657	14.75 ± 23.21
Average time of single lead extraction	[minutes]	8.19 ± 10.76 *p* = 0.203	8.07 ± 9.28 *p* = 0.781	8.72 ± 9.72 *p* = 0.669	8.36 ± 10.20 *p* = 0.214	9.68 ± 14.00
Technical problem during TLE (any)	n (%)	20 (14.49) *p* = 0.044	6 (18.18) *p* = 0.737	18 (20.22) *p* = 0.746	44 (16.92) *p* = 0.063	476 (22.16)
Need to change venous approach	n (%)	1 (0.72) *p* = 0.171	2 (6.06) *p* = 0.661	5 (5.62) *p* = 0.331	8 (3.08) *p* = 0.912	68 (3.17)
Lead-to-lead scarring	n (%)	6 (4.35) *p* = 0.318	2 (6.06) *p* = 0.882	5 (5.62) *p* = 0.789	13 (5.00) *p* = 0.295	149 (6.94)
Fracture of extracted lead	n (%)	2 (1.45) *p* = 0.034	0 (0.00) *p* = 0.259	6 (6.74) *p* = 0.002	8 (3.08 *p* = 0.151	134 (6.24)
Byrd dilator collapse/torsion	n (%)	5 (3.62) *p* = 0.841	2 (6.06) *p* = 0.799	3 (3.37) *p* = 0.891	10 (3.85) *p* = 0.970	79 (3.69)
Obstruction at lead entry site	n (%)	12 (8.70) *p* = 0.912	3 (9.09) *p* = 0.801	6 (6.74) *p* = 0.628	21 (8.08) *p* = 0.782	186 (8.66)
Two or more technical problems	n (%)	8 (5.80) *p* = 0.769	2 (6.06) *p* = 0.933	2 (2.25) *p* = 0.382	12 (4.62) *p* = 0.994	104 (4.84)
**Use of additional tools**						
Evolution (old and R-L) or TightRail	n (%)	0 (0.00) *p* = 0.278	0 (0.00) *p* = 0.935	0 (0.00) *p* = 0.410	0 (0.00) *p* = 0.062	37 (1.72)
Metal sheath	n (%)	11 (7.97) *p* = 0.977	3 (9.09) *p* = 0.858	5 (5.62) *p* = 0.378	19 (7.31) *p* = 0.618	181 (8.43)
Lasso catheter/snare	n (%)	2 (1.45) *p* = 0.172	1 (3.03) *p* = 0.914	4 (4.49) *p* = 0.897	7 (2.69) *p* = 0.321	90 (4.19)
Basket catheter	n (%)	0 (0.00) *p* = 0.399	0 (0.00) *p* = 0.835	0 (0.00) *p* = 0.616	0 (0.00) *p* = 0.178	24 (1.12)
Temporary pacing during the procedure	n (%)	40 (28.97) *p <* 0.001	12 (36.36) *p* = 0.601	18 (20.22) *p* = 0.196	70 (26.92) *p <* 0.001	315 (14.66)
**New lead(s) implantation**						
Reestablished vein access (using lead extraction)	n (%)	58 (42.028)	19 (57.576)	32 (35.955)	109 (41.923)	518 (24.154)
Maintained venous approach (using lead extraction)	n (%)	75 (54.347)	11 (33.333)	49 (55.056)	135 (51.923)	1277 (59.451)
New insertion site parallel to the existing lead	n (%)	5 (3.623)	3 (9.091)	8 (8.989)	16 (6.154)	353 (16.433)

TLE—transvenous lead extraction, CRTD—cardiac resynchronization cardioverter defibrillator, CRT-P—cardiac resynchronization pacemaker, ICD—implantable cardioverter defibrillator, ICD-V—one chamber ICD, ICD-D—dual chamber ICD, SD—standard deviation, N/[n]—number. The control group 2 served as a comparison group in the statistical evaluation.

**Table 4 ijerph-20-00291-t004:** Lead management strategy, TLE outcomes, and long-term mortality.

		Upgrade to CRT-D	Upgrade to CRT-P	Upgrade to ICD	All Upgrades (to CRT-D, CRT-P and to ICD-V or ICD-D)	TLE for Other Non-Infectious Indications without Upgrades
**Group**		**1a**	**1b**	**1c**	**1**	**2**
**Number of Patients**		**138**	**33**	**89**	**260**	**2148**
		**n (%)**	**n (%)**	**n (%)**	**n (%)**	**n (%)**
**Lead management strategy**						
All leads were extracted	n (%)	94 (68.12) *p* = 0.648	25 (75.76) *p* = 0.313	68 (76.40) *p* = 0.059	187 (71.92) *p* = 0.058	1414 (65.83)
Functional lead was left in place	n (%)	44 (31.88) *p* = 0.789	8 (24.24) *p* = 0.358	18 (20.22) *p* = 0.013	70 (26.92) *p* = 0.043	717 (33.38)
Non-functional lead was left behind	n (%)	0 (0.00) *p* = 0.740	0 (0.00) *p* = 0.490	1 (1.12) *p* = 0.938	1 (0.38) *p* = 0.992	13 (0.61)
Non-functional superfluous lead was extracted	n (%)	5 (3.62) *p* = 0.075	2 (6.06) *p* = 0.894	10 (11.24) *p* = 0.421	17 (6.54) *p* = 0.406	177 (8.24)
**TLE efficacy and complications**						
Major complication (any)	n (%)	0 (0.00) *p* = 0.010	0 (0.00) *p* = 0.690	0 (0.00) *p* = 0.225	0 (0.00) *p* = 0.013	57 (2.33)
Hemopericardium	n (%)	0 (0.00) *p* = 0.234	0 (0.00) *p* = 0.945	0 (0.00) *p* = 0.418	0 (0.00) *p* = 0.065	36 (1.68)
Hemothorax	n (%)	0 (0.00) *p* = 0.587	0 (0.00) *p* = 0.072	0 (0.00) *p* = 0.383	0 (0.00) *p* = 0.913	4 (0.19)
Tricuspid valve damage during TLE (severe)	n (%)	0 (0.00) *p* = 0.698	0 (0.00) *p* = 0.527	0 (0.00) *p* = 0.938	0 (0.00) *p* = 0.382	14 (0.65
Emergent cardiac surgery	n (%)	0 (0.00) *p* = 0.326	0 (0.00) *p* = 0.925	0 (0.00) *p* = 0.532	0 (0.00) *p* = 0.113	29 (1.35)
Death: procedure-related (intra-, post-procedural)	n (%)	0 (0.00) *p* = 0.710	0 (0.00) *p* = 0.120	0 (0.00) *p* = 0.490	0 (0.00) *p* = 0.954	5 (0.23)
Death: indication-related (intra-, post-procedural	n (%)	0 (0.00) MN	0 (0.00) MN	0 (0.00) MN	0 (0.00) MN	0 (0.00)
partial radiographic success (retained tip or <4 cm lead fragment)	n (%)	2 (1.45) *p* = 0.179	0 (0.00) *p* = 0.453	3 (3.37) *p* = 0.930	5 (1.92) *p* = 0.115	89 (4.14)
Complete clinical success	n (%)	138 (100.0) *p* = 0.131	33 (100.0) *p* = 0.764	89 (100.0) *p* = 0.276	260 (100.0) *p* = 0.024	2098 (97.67)
Complete procedural success	n (%)	136 (98.55) *p* = 0.096	33 (100.0) *p* = 0.368	86 (96.63) *p* = 0.674	255 (98.08) *p* = 0.042	2042 (95.07)
**Mortality after TLE**						
Alive (survival rate) after a mean follow-up of 4.81 ± 3.44 years (1–5394 days)	n (%)	81 (58.70) *p <* 0.001	25 (75.76) *p <* 0.001	60 (67.42) *p <* 0.001	166 (63.85) *p <* 0.001	1615 (75.19)
First two-day mortality (first 48 h)	n (%)	0 (0.00) *p* = 0.587	0 (0.00) *p* = 0.072	0 (0.00) *p* = 0.383	0 (0.00) *p* = 0.913	4 (0.19)
1-month mortality after TLE; 2–30 days n (% of patients with follow-up longer than 2 days)	n (%)	3/138 (2.17) *p* = 0.063	0/33 (0.00) *p* = 0.409	0/89 (0.00) *p* = 0.932	3/(1.15) *p* = 0.393	11/2137 (0.51)
1-year mortality after TLE (31–365 days); n (% of patients with follow-up longer than 30 days)	n (%)	8/129 (6.20) *p* = 0.494	2/32 (6.25) *p* = 0.963	10/82 (12.20) *p* = 0.006	20/246 (8.13) *p* = 0.023	91/2063 (4.24)
3-year mortality after TLE (366–1095 days); n (% of patients with follow-up longer than 365 days	n (%)	20/114 (17.54) *p <* 0.001	1/25 (4.00) *p* = 0.813	6/70 (8.57) *p* = 0.938	27/211 (12.80) *p* = 0.023	138/1839 (7.50)
Death at >3 years after TLE (after 1095 days); n (% of patients with follow-up longer than 1095 days)	n (%)	26/114 (22.81) *p* = 0.839	5/16 (31.25) *p* = 0.641	13/56 (32.21) *p* = 0.901	44/128 (34.38) *p* = 0.012	289/1362 (21.22)
All deaths	n (%)	57 (41.30) *p <* 0.001	8 (24.24) *p* = 0.898	29 (32.58) *p* = 0.126	94 (36.15) *p <* 0.001	533 (4.81)

TLE—transvenous lead extraction, CRTD—cardiac resynchronization cardioverter defibrillator, CRT-P—cardiac resynchronization pacemaker, ICD—implantable cardioverter defibrillator, ICD-V—one chamber ICD, ICD-D—dual chamber ICD, SD—standard deviation, N/[n]—number, MN—methodically noncomparable. The control group 2 served as a comparison group in the statistical evaluation.

## Data Availability

Readers can access data supporting the conclusions of the study upon reasonable request to the authors.

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
