# Peer review of "Efficacy and Safety of Transvenous Lead Extraction at the Time of Upgrade from Pacemakers to Cardioverter-Defibrillators and Cardiac Resynchronization Therapy"

_ijerph, 2022, doi:10.3390/ijerph20010291_

Round 1

Reviewer 1 Report

The grammar needs a review prior to publication 

The fact that they succeeded in 100% cases in interesting.. did the authors not encounter cases where the leads pulled through prior to gaining access?

Reviewer 2 Report

First of all, English style must be improved. It should be checked by a native speaker. Now, it seems like the Google translate tool was used to translate from Polish to English.

Commas should be replaced by dots in the numbers across whole manuscript. 

The aim is not clearly written in the abstract. 

I do not understand what does it mean to "rebuild venous approach" which was used many times in the manuscript. I suppose the authors wanted to express "retain venous access" but this is not clear for me. 

I do not understand why the authors use word "up-grading" insted of "upgrade".

I do not understand why the authors introduced numbers for analysed grups: 1-5. I think it could be omitted since the numbers are not used in the text. If necessary, the authors could use number 1 for all upgrade procedures, number 2 for control groups, and 1a-1c for subgroup analysis.  

I do not understand why there are three digits after comma in the table ie. 28,23+-7,959, etc. 

In the methods section the definitions of moderate, severe and complete venous stenosis were not explained.

There are many minor languege/spelling errors i.e. "1our", "underlaying", "TLE complicity", "whit", missing brackets, etc. 

The tables are not easy to follow. It should be clearly stated in the table that other noninfective TLE patients are the control group. Probably, the tables should be split into all upgrades vs. other TLE and subgroup analysis (specific upgrades vs control group).

Dwell time, lead duration should be expressed in years for clarity, not in months. 

Days of follow-up should be replaced by the years of follow-up. 

I do not understand why the authors created figure 2 showing the role of TEE during TLE procedures, since there is nothing about TEE in the methods or how often was the TEE used in the procedures. The results do not support the use of TEE. 

The mortality of patients upgraded to CRT is obviously lower than mortality of other TLE patients, which is related to higher comorbidity index. It would be interesting to see an analysis comparing upgrades procedures to ICD/CRT with TLE vs upgrades without TLE.

Round 2

Reviewer 2 Report

Response to Reviewer 2 Comments

1.First of all, English style must be improved. It should be checked by a native speaker. Now, it seems like the Google translate tool was used to translate from Polish to English.

Answer: The manuscript has been reviewed by a native speaker and appropriate changes have been made

OK

2.Commas should be replaced by dots in the numbers across whole manuscript.

Answer: Done. 

OK

3.The aim is not clearly written in the abstract.

Answer: The aim of the study was added to Abstract 

This is still unclear for a reader. 

“TLE is considered riskier in patients with multiple diseases. This study set out to critically evaluate the findings.” was added. In my opinion this is still not easy to understand what the authors did. I cannot accept abstract in which reader is forced to think what did Authors want to express. Wouldn’t be easier for a reader to write that aim of the study was to compare 

Please find the examples from TLE publications where aim is stated (examples from other publications of Authors):

“Thus, this study determined the differences and specific characteristics of TLE in children vs. adults.”

“The aim of study was to explore the frequency of repeat TLE, its safety, predisposing factors, as well as effectiveness of repeat procedures.”

“Assessment of the impact of the organization of TLE on the safety of procedures.”

“We aimed to compare procedure complexity and the incidence of the TLE major complications (MC) in groups where extracted leads were under 10 years, 10-20 years, 20-30 years (old) and over 30 years (very old).”

Additionally, the abstract is now 26 words too long to fulfill journal submission criteria. 

4.I do not understand what does it mean to "rebuild venous approach" which was used many times in the manuscript. I suppose the authors wanted to express "retain venous access" but this is not clear for me. 

Answer: In patients with complete venous occlusion, lead extraction helps reestablish access to an occluded vein. The released lead is removed through a dissecting catheter (Byrd dilator, Evolution, TightRail) and the deflated catheter allows insertion of one or more guidewires into the RA or preferably IVC (Figure 1 and Figure 3). The sheath dilator is removed, and the guidewires remain in the vascular system for implantation of a new lead or leads. For clarity, the word “rebuild” has been replaced with “reestablish” to describe the possibility of implanting a new lead despite complete venous occlusion.

In patients with partial venous obstruction (mild or moderate), implantation of a new lead is possible by creating a new venous access. However, if the lead is extracted using a dissecting catheter, a guidewire can be introduced through the catheter, enabling implantation of the new lead, without puncture of the axillary or subclavian vein. We refer to this situation as maintained venous approach.

In short: Reestablished vein access - in case of complete or severe venous obstruction. Maintained venous approach – in case of preserved venous patency to avoid a new puncture in the axillary or subclavian vein.

There is no such term as “reestablish vein access”. I find it unnecessary to create a new medical term when better equivalent is present (to restore venous access)

OK

5.I do not understand why the authors use word "up-grading" instead of "upgrade".

Answer: It was corrected. Thank you.

OK

6. I do not understand why the authors introduced numbers for analysed grups: 1-5. I think it could be omitted since the numbers are not used in the text. If necessary, the authors could use number 1 for all upgrade procedures, number 2 for control groups, and 1a-1c for subgroup analysis.  

Answer: OK, it was done

OK

7.I do not understand why there are three digits after comma in the table ie. 28,23+-7,959, etc. 

Answer: Mean and SD values are presented in a uniform manner as numbers with two decimal places. Thanks for noticing inconsistency

OK

8. In the methods section the definitions of moderate, severe and complete venous stenosis were not explained.

Answer: Venous patency before lead extraction defined as maximal narrowing and the number of affected veins is summarized in table 2. We've added this information to the Methods section

OK

9.There are many minor languege/spelling errors i.e. "1our", "underlaying", "TLE complicity", "whit", missing brackets, etc. 

Answer: All minor language/spelling errors have been corrected.

OK

10.The tables are not easy to follow. It should be clearly stated in the table that other noninfective TLE patients are the control group. Probably, the tables should be split into all upgrades vs. other TLE and subgroup analysis (specific upgrades vs control group).

Answer: We have renumbered the groups. The upgrade group (1) and three subgroups (1a, 1b and 1c) are compared with the control group (2) (TLE for other non-infectious indications without upgrade). Below each table we have added an explanation that the control group 2 served as a comparison group in statistical evaluation

OK

11.Dwell time, lead duration should be expressed in years for clarity, not in months. 

Answer: It has been done. Now implant duration is expressed in years.

OK

12.Days of follow-up should be replaced by the years of follow-up. 

Answer: It has been done. Days of follow-up have been replaced by years.

OK

 13.I do not understand why the authors created figure 2 showing the role of TEE during TLE procedures, since there is nothing about TEE in the methods or how often was the TEE used in the procedures. The results do not support the use of TEE. 

Answer: Information on TEE for monitoring patients undergoing TLE has been added to the Methods section and the issue has been raised in the Discussion by referring to Figure 2 and the newly cited references.

OK

 14.The mortality of patients upgraded to CRT is obviously lower than mortality of other TLE patients, which is related to higher comorbidity index. It would be interesting to see an analysis comparing upgrades procedures to ICD/CRT with TLE vs upgrades without TLE.

Answer: Yes, of course, the differences are clear if you compare the group of upgraded patients with other patients, in whom infections are often the main indication for TLE. In the current study, we excluded patients with infections and analyzed only patients without CIED infections. Patients upgraded to ICD and CRT-P/CRT-D were slightly older, had significantly lower EF, more frequent IHD, heart failure, renal failure, and higher Charlson comorbidity index. For these reasons, mortality in this group was slightly higher than in the control group, which proves the clinical effectiveness of upgrade procedures performed despite varying degrees of venous obstruction.

Our TLE database currently includes 3,800 procedures performed over the last 17 years. Unfortunately, we do not have a comparable patient base with CIED-related procedures but without TLE.

OK
